# Rare Human Missense Variants can affect the Function of Disease-Relevant Proteins by Loss and Gain of Peroxisomal Targeting Motifs

**DOI:** 10.3390/ijms20184609

**Published:** 2019-09-17

**Authors:** Cheng-Shoong Chong, Markus Kunze, Bernhard Hochreiter, Martin Krenn, Johannes Berger, Sebastian Maurer-Stroh

**Affiliations:** 1Bioinformatics Institute, Agency for Science, Technology and Research (A*STAR), Singapore 138671, Singapore; e0161111@u.nus.edu; 2National University of Singapore Graduate School for Integrative Sciences and Engineering (NGS), National University of Singapore, Singapore 119077, Singapore; 3Medical University of Vienna, Center for Brain Research, Department of Pathobiology of the Nervous System, 1090 Vienna, Austria; 4Medical University of Vienna, Center for Physiology and Pharmacology, Institute for Vascular Biology and Thrombosis Research, 1090 Vienna, Austria; 5Department of Neurology, Medical University of Vienna, 1090 Vienna, Austria; 6Institute of Human Genetics, Technical University Munich, 81675 Munich, Germany; 7Department of Biological Sciences, National University of Singapore, Singapore 117558, Singapore; 8Innovations in Food and Chemical Safety Programme (IFCS), Agency for Science, Technology and Research (A*STAR), Singapore 138671, Singapore

**Keywords:** peroxisomal targeting signal 1, PEX5, peroxisome, protein transport, missense variant, mistargeting, gnomAD, disease

## Abstract

Single nucleotide variants (SNVs) resulting in amino acid substitutions (i.e., missense variants) can affect protein localization by changing or creating new targeting signals. Here, we studied the potential of naturally occurring SNVs from the Genome Aggregation Database (gnomAD) to result in the loss of an existing peroxisomal targeting signal 1 (PTS1) or gain of a novel PTS1 leading to mistargeting of cytosolic proteins to peroxisomes. Filtering down from 32,985 SNVs resulting in missense mutations within the C-terminal tripeptide of 23,064 human proteins, based on gene annotation data and computational prediction, we selected six SNVs for experimental testing of loss of function (LoF) of the PTS1 motif and five SNVs in cytosolic proteins for gain in PTS1-mediated peroxisome import (GoF). Experimental verification by immunofluorescence microscopy for subcellular localization and FRET affinity measurements for interaction with the receptor PEX5 demonstrated that five of the six predicted LoF SNVs resulted in loss of the PTS1 motif while three of five predicted GoF SNVs resulted in *de novo* PTS1 generation. Overall, we showed that a complementary approach incorporating bioinformatics methods and experimental testing was successful in identifying SNVs capable of altering peroxisome protein import, which may have implications in human disease.

## 1. Introduction 

Inter-individual DNA sequence variations in the human genome provoke diverse consequences ranging from physical trait differences to altered disease susceptibility and even drug therapy failure [1,2,3]. Non-synonymous single nucleotide variants (SNVs) resulting in amino acid substitutions or missense variants have been implicated in many human genetic diseases [4]. To date, more than 100 million validated SNVs are listed in the Single Nucleotide Polymorphism Database (dbSNP build 151) [5], and with the advent of inexpensive next-generation sequencing technology, many more novel and rare SNVs are being discovered [6].

While the ability of missense variants to cause diseases is clear, their effects on the protein can be complex and manifest in several ways. At the protein level, they have been found to cause protein instability, change protein flexibility, abrogate protein–macromolecular and protein–chemical interactions, reduce enzyme activity, and even modify protein function [7,8,9]. Importantly, such SNVs can also lead to aberrant protein localization in the cell by inactivating targeting signals [10,11], although such effects have hardly been investigated since most targeting signals initiating these translocation processes are rather robust against individual mutations.

In this study, we focused on SNVs leading to the generation or destruction of peroxisomal targeting signal type-1 (PTS1) and their potential impact on sorting of proteins into the peroxisomes. Peroxisomes are single membrane bound organelles housing a variety of metabolic reactions that degrade various fatty acids through β-oxidation, break down hydrogen peroxide and other reactive oxygen species (ROS), and catabolize D-amino acids [12]. In humans, peroxisomes also contribute to the biosynthesis of ether-phospholipids, including plasmalogens or bile acids [13]. The importance of peroxisomes for human physiology is highlighted in many inherited human diseases caused by a complete dysfunction of peroxisomes (Zellweger syndrome spectrum) [14,15] or by the lack of an individual enzyme or transporter protein [16].

Soluble proteins destined for the peroxisomal lumen are equipped with a peroxisomal targeting signal which resides either at the extreme C-terminus (PTS1) or close to the N-terminus (PTS2) [13]. PTS1 motifs have been characterized as a C-terminal tripeptide Ser-Lys-Leu (SKL) or conserved variants thereof ([S/A/C]–[K/R/H]-L-^COO−^]) [17,18], although later, a much broader number of tripeptides were found to act as targeting signals. Moreover, the sequence preceding the C-terminal tripeptide, the upstream sequence, modulates the binding strength to the PTS1 receptor PEX5 [19,20]. Mistargeting of proteins has been linked to diseases in the context of arbitrary generation of alternative targeting signals for peroxisomal proteins [21,22,23]. Furthermore, cytosolic proteins should be sensitive to mislocalization because the *de novo* generation of a targeting signal should induce transport, and thus, it should deplete the protein from the cytosol. In this respect, the PTS1 signal is a good candidate for study as it is (i) exposed to the extreme C-terminal end, (ii) reliably predictable [24], and (iii) can be modulated by individual point mutations, whereas other targeting signals are rather tolerant against mutations, but also hard to induce.

However, much is still unknown about the potential of SNVs in affecting protein import into peroxisomes by gain or loss of PTS1, although computational algorithms and tools [25,26] including integrative methods [27] have been developed to aid in delineating between SNVs with and without functional consequences. Using a combination of such tools and along with experimental validation of PTS1 signal quality, we performed a systematic analysis of SNVs affecting protein transport into peroxisomes after mining the entire Genome Aggregation Database (gnomAD) [28] for relevant missense variants occurring in the extreme C-termini of all human proteins. At the first level, we identified loss of function variants (LoF) inactivating PTS1 in known, disease-relevant peroxisomal proteins. These LoF mutants are expected to ablate the proper localization of these enzymes, and thereby interrupting peroxisomal metabolism similar to a loss of the enzymatic activity. In a second step, we searched for SNVs introducing a PTS1 motif in a cytosolic protein, which was expected to induce peroxisomal import of the protein, thereby functionally depleting the cytosol of this protein. To the best of our knowledge, these are the first descriptions of the ability of missense variants occurring naturally in the human population to abolish or generate a PTS1 signal and consequently alter the localization of the affected protein into or out of the peroxisome. Our findings would be important in the context of analyzing the effects of variant-induced aberrant localization on protein function, and reinforces the need to evaluate targeting signal changes when determining the disease-relevance of a protein mutation amongst other factors.

## 2. Results

### 2.1. Mining of gnomAD for SNVs Causing Missense Mutations in C-terminal Tripeptides

From a core set of manually curated protein coding regions from the Consensus Coding Sequence project (CCDS), we obtained protein and DNA sequence information along with protein coding genomic coordinates for 30,539 proteins and their isoforms. After exclusion of 1642 CCDS-withdrawn proteins, we applied a chromosome and position-based query of 125,748 whole exome and 15,708 whole genome sequencing data from gnomAD (see Method 4.1) to filter out SNVs located at the last three codons preceding the stop codon of the gene transcripts encoding the remaining proteins. gnomAD is a sequence variation database containing harmonized variant data from more than 140,000 human samples collected from a broad range of studies (full list available at https://gnomad.broadinstitute.org/about). We chose this database for our analysis as it is currently the largest publicly available human sequence variation database and the data is also easily accessible.

Additionally, we chose to focus on SNVs lying in the last three codons of each transcript as they can potentially generate or abolish a PTS1 mediating the interaction with the PTS1 receptor (PEX5) by mutating the C-terminal tripeptide of the analyzed proteins. The importance of this tripeptide to PTS1-mediated peroxisome targeting has been extensively studied compared to its upstream sequence, and hence, it is well suited to predict [25] and proof functional changes in the PTS1. Figure 1 depicts the location of the tripeptide motif in PTS1 in a previously crystallized complex (pdb ID: 2c0l) that consists of the TPR region of PEX5 and its PTS1-containing ligand (human SCP2).

In this region, we found that 23,064 protein encoding transcripts covering 15,180 genes contained SNVs (Figure 2A). Given that not all SNVs cause non-synonymous amino acid changes, we only retained missense variants, and in total, we attained 32,985 bi- or multi-allelic SNVs capable of mutating one of the last three amino acids of these proteins. Considering the entire gene (*n* = 18,592) and protein set analyzed (*n* = 28,897), approximately 80% of the proteins and 81.6% of genes had at least one missense variant from at least one human sample in gnomAD capable of mutating the C-terminal tripeptides under study.

### 2.2. Investigation of SNVs Resulting in PTS1 Ablation (Loss-of-Function)

#### 2.2.1. Identification and Prioritization of SNVs Resulting in Loss-of-Function PTS1 for Experimental Testing

From curation of the peroxisomal databases [30] and our literature survey, we identified 35 genes encoding PTS1-containing human peroxisomal proteins (Figure 2B, Appendix A). To focus on studying potential mistargeting in disease-relevant genes, we narrowed the list to six genes that had been broadly linked to at least one human disease in the Online Mendelian Inheritance in Man (OMIM) database. These gene products must also be localized solely in the peroxisome based on UniProt subcellular localization annotations (see Section 4.3, Section 4.5). After filtering for missense variants that passed gnomAD associated quality filters and that affected proteins with a traditional PTS1 consensus motif [18], six SNVs in four genes remained to be tested for motif inactivation (Table 1, Figure 2B).

#### 2.2.2. Experimental Investigation of Functional Consequences of SNVs in PTS1 Motifs

Next, we experimentally verified the functional consequences of the selected SNVs. These SNVs changed the C-terminal tripeptide from typical PTS1 motifs to derivatives thereof, which were not covered by the original suggested PTS1 consensus sequence of the C-terminal tripeptide [18]. To verify the effect of these point mutations on the quality of the PTS1, we first generated EGFP fusion proteins extended by the last 12 amino acids of the native enzymes or with variants thereof harboring the mutated C-terminal tripeptide. When these proteins were expressed in COS7 cells, we found that all EGFP-proteins extended by the PTS1 motifs of the native peroxisomal proteins (HSD17B4, DAO, ACOX2, and EHHADH) were found in punctate structures colocalizing with the peroxisomal marker PMP70, which confirmed the functionality of these PTS1 motifs (Figure 3A,D,F,H). In contrast, fusion proteins encoding the same PTS1, but harboring the point mutations HSD17B4-K760E (Figure 3B), DAO-S345F (Figure 3E), and ACOX2-K680T (Figure 3G) appeared exclusively cytosolic indicating an inactivation of the PTS1, whereas HSD17B4-A759P (Figure 3C) was clearly punctate suggesting that the functionality of the PTS1 was hardly affected. In contrast, the variant EHHADH-S721G was predominantly cytosolic (Figure 3I) and the variant EHHADH-S721N was nearly exclusively cytosolic with a faint peroxisomal background in a fraction of cells (Figure 3J).

### 2.3. Investigation of SNVs Resulting de novo PTS1 Generation in Cytosolic Proteins (Gain-of-Function)

#### 2.3.1. Identification and Prioritization of Gain-of-Function SNVs Generating *de novo* PTS1 for Experimental Testing

From the entire collection of 32,985 SNVs described above, we concurrently applied three filters that were based on the annotation of subcellular localization data, on variant quality, and on the evaluation by the PTS1 predictor [25], to search for SNVs with the potential to generate a *de novo* PTS1 at the C-terminus of a cytosolic protein (see Methods 4.5, Figure 2B). The strict filters identified 16 SNVs in 22 CCDS transcripts of 16 genes (Appendix A). Among these, we chose five SNVs which generated PTS1-like tripeptides (MT) from non-PTS1 WT tripeptides at the C-termini of reasonably long proteins (<1000 a.a) without transmembrane domains. (Figure 2B, Table 2).

#### 2.3.2. Experimental Validation of Gain-of-function SNVs Leading to *de novo* Generation of PTS1

Similarly, we generated expression plasmids for EGFP extended either by the C-terminus of the native proteins or by the mutated form. When the EGFP fusion proteins encoding the native C-terminus of these proteins (PPP4R4, RFLNA, ARHGAP1, HPGDS, and GLTP) were expressed in COS7 cells and the subcellular localization of the reporter protein was determined by fluorescence microscopy, we found that EGFP was evenly distributed across the cytosol for all (Figure 4A,C,E,I), except for HPGDS. This result showed punctate staining in front of a strong background, and these punctate structures colocalized with PMP70 (Figure 4G) suggesting that four out of five of these C-terminal ends were non-functional, whereas the C-terminus of HPGDS appeared to act as weak PTS1. However, when the variants of the C-terminal sequences encoding the point mutation were expressed, we found that the fusion proteins encoding the C-terminus of the mutants PPP4R4-P873L (Figure 4B), RFLNA-T134K (Figure 4D), and HPGDS-T197S (Figure 4H) showed a clear punctate staining indicating the functionality of the modified C-terminus as PTS1. Moreover, the EGFP-fusion protein with the C-terminus of ARHGAP1-G438R presented with a weak peroxisomal staining (Figure 4F), and that of GLTP-Y207C appeared completely cytosolic (Figure 4J). These results confirmed the ability of SNVs to generate novel functional PTS1 variants *de novo*. Moreover, we found that in one case the native C-terminus had a weak ability to mediate peroxisomal import (HPGDS), whereas, in another, the mutation did not generate a functional PTS1 (GLTP). However, in this case, the mutation increased the numerical PTS1-score only slightly, although this was sufficient to cross a threshold and change the qualitative prediction from “Twilight zone” to “Targeted.” The inability of this GLTP mutant to act as a PTS1 might be due to the valine being the last residue of the tripeptide as this amino acid has been shown to weaken known PTS1 motifs [31].

### 2.4. Measurement of Relative Affinity Before and After Point Mutations for both GoF and LoF Mutants

To retrace the effect of SNVs in the last three amino acids more closely to changes in the binding strength to the PTS1-receptor PEX5, we took advantage of a recently developed novel approach [32] to determine the relative binding affinity of PEX5 to diverse PTS1 motifs. Subjecting the abovementioned peptides to this FRET-based measurement of protein–protein interactions in living cells, we were able to determine the change in the affinity to PEX5 caused by individual point mutations for each PTS1 (Figure 5).

We found that for all point mutations reflecting SNVs that inactivated the PTS1, the affinity to PEX5 was also drastically reduced (HSD17B4-K760E, DAO-S345F, ACOX2-K680T, EHHADH-S721N) just as in the case of slight residual import (EHHADH-S721G) (Figure 5A). In contrast, hardly any change was observed for the SNV that did not ablate peroxisomal import (HSD17B4-A759P). This outcome confirmed the results of our previous experiments using an orthogonal method and showed that the functionality of the PTS1 correlates with its affinity to PEX5. Similarly, all SNVs that converted a C-terminal end into a functional PTS1 also increased the affinity of the peptide to PEX5 (PPP4R4-P873L, RFLNA-T134K, ARHGAP1-G438R) although to different extents, whereas the mutation that did not act as PTS1 did not interact with PEX5 (GLTP-Y207C) (Figure 5B). Finally, for HPGDS, the C-terminal sequence in the native form not only enabled peroxisomal import but it mediated interaction with PEX5, although the SNV (HPGDS-T197S) presented with an even higher affinity.

### 2.5. Assessment of PTS1 Predictor and FoldX Predictions of PTS1 Signal Quality

Given that we used computational predictions such as the PTS1-predictor to identify SNVs that could affect peroxisomal targeting and experimentally determined the effect of the SNVs, we could use the experimental data generated in this study as an independent validation of the performance of such tools for future applications. Here, we assessed the overall performance of the PTS1-predictor and another predictor that we had used previously (structural binding prediction using FoldX) [33] by comparing the predictions with the experimental results (immunofluorescence microscopic analysis (IFA) and FRET affinity measurements) (Table 3).

FoldX is an empirical force field that can be used to predict a change in interaction energy between the ligand and receptor upon one or more amino acid mutations at their interaction interface. Positive ddG_bind_ values denote a reduction in binding strength, and conversely, a negative value suggests an improved interaction. Based on the results in Table 3, seven of the 11 point mutations were predicted to destabilize the PTS1–PEX5 interaction while the rest of the point mutants were predicted to stabilize further the interaction between the two binding partners. Specifically, for the LoF mutants that were confirmed by IFA, FoldX correctly predicted ablation of PTS1 for five of the six variants as most of PTS1 generated were nearly or completely cytosolic after introduction of the point mutant. For HSD17B4 A759P which retained peroxisome localization even after mutation, FoldX predicted a slight ablated interaction (ddG_bind_ = +0.154 kcal/mol) between the HSD17B4 mutant and PTS1 receptor (PEX5) which was not a wrong prediction given the level of significance for FoldX energy changes is ±0.5 kcal/mol.

At the same time, of the five GoF related SNVs, FoldX projected that three would lead to higher interaction affinity and thus, the peptide can act as PTS1. In comparison to the IFA results where all point mutants generated PTS1 signals except for GLTP Y207C, we observed that two (RLFNA and ARHGAP1) missense variants were correctly predicted to stabilize the interaction. In total, FoldX correctly predicted eight of the 11 SNVs to change affinity to PEX5 resulting in a change in localization. Further comparison of the absolute ddG_bind_ values with the FRET affinity measurements using correlation analysis (Figure 6A) revealed that the ddG_bind_ values had a statistically significant correlation with the log_10_ K_a_^app^ ratio of the affinity measurements (Pearson’s *r* = 0.68, *p* = 0.02).

The PTS1-predictor is a prediction tool that assesses PTS1 quality using a position-specific scoring matrix developed from a database of known PEX5 binding PTS1-containing proteins and peptides from two hybrid assays. For an input consisting of the primary sequence of a protein, the algorithm categorizes the protein into three categories: non-peroxisomal (“Not Targeted”), peroxisomal (“Targeted”), and intermediate between the two categories (“Twilight zone”).

For the LoF mutants, the tool predicted that half of the six SNVs would cause a loss in peroxisome targeting from “Twilight zone” to “Not targeted” or “Targeted” to “Twilight zone”/”Not targeted.” Conversely, the other three variants were predicted to have no change in localization even after point mutation. When compared to the IFA results, four of these predictions were in concordance while the other two EHHADH gene variants were incorrectly predicted to have no change in localization. Concurrently, the GoF-associated variants were all predicted to change from being “Not targeted” or “Twilight zone” to being “Targeted” to the peroxisome after point mutation (Table 2). Three of these five predictions were true based on the IFA results (ARHGAP1, RFLNA, PPP4R4 gene variants), while for the other two variants no change in localization was observed. Altogether, the prediction tool correctly predicted localization change in seven of the 11 SNVs tested when considering the crude output labels. A more detailed analysis of the absolute score values from the algorithm was also performed, where we computed the score difference between the WT variant and the point mutant (Table 3) and correlated this with the FRET affinity measurements similar to the FoldX analysis. A statistically highly significant negative correlation between the score difference and the experimental measurements (log_10_ K_a_^app^ ratio, see Method 4.7) was obtained (Pearson’s *r* = −0.93, *p* < 0.001) suggesting that as relative affinity of the WT over the MT variant increases (log ratio will increase), the predicted score difference decreases (Figure 6B). Since the method issequence-based and the calculation is fast over a larger number of sequences (<1 sec per sequence), it would be well suited for high-throughput screening of effects of SNVs on PTS1 motifs. Additionally, in large datasets the calculated score differences could serve as a finer differentiation point for mutations resulting in the same pairs of categorical outcomes (e.g., “Targeted” -> “Not targeted”).

### 2.6. Disease-Relevance of Investigated SNVs as Consequence of Abolishment or Gain in Peroxisome Targeting

To evaluate disease relevance of the selected SNVs, we used the gnomAD database (comprising >140,000 control individuals) to assess the minor allele frequencies (MAF) of all investigated SNVs in addition to obtaining in silico predictions from variant pathogenicity prediction tools, such as SIFT and Polyphen-2. While no homozygous carriers were listed for any SNVs, MAFs ranged from singletons (one allele in gnomAD) to a maximum of 0.028% (79 alleles in gnomAD). To account for potential existing links to human diseases, we screened publicly available databases that provide evidence for associations between genetic variants and human diseases. In the ClinVar database, we found the most frequent variant S721G (in EHHADH) to be classified as “likely benign” while all other SNVs were not listed. Collectively in the entire dataset of over 30,000 SNVs, 29 of them were associated with at least one “pathogenic” or “likely pathogenic” clinical significance in this database. Of note, only four of the genes have so far been linked to human diseases according to the Online Mendelian Inheritance in Man (OMIM) database. To date, there is only limited evidence for disease associations of DAO (schizophrenia) [34,35,36], RFLNA (spondylocarpotarsal synostosis) [37], and EHHADH (Fanconi renotubular syndrome) [21,22]. However, homozygous variants in HSD17B4 (D-bifunctional protein deficiency (MIM# 261515), Perrault syndrome 1 (MIM# 233400)) and ACOX2 (Bile acid synthesis defect, congenital (MIM# 617308)) have reliably been associated with monogenic diseases. In contrast, no human diseases have so far been linked to the genes PPP4R4, ARHGAP1, HPGDS, and GLTP according to OMIM. Independent of known disease associations, effects of the studied SNVs could be detrimental for the respective protein’s molecular and cellular functions as discussed below.

For the D-bifunctional enzyme (HSD17B4) which is associated with severe genetic disease (Table 4), inactivation of the PTS1 creates a purely cytosolic enzyme that cannot exert its peroxisomal function of fatty acid degradation and bile acid side chain shortening [38]. Therefore, this mutation should mimic an inactive enzyme with a recessive Mendelian inheritance.

Acyl-CoA oxidase 2 (ACOX2) is an enzyme that mediates FAD-dependent dehydrogenation of the side chain of bile acids [39]. The lack of peroxisomal ACOX2 due to the SNV inactivating the PTS1 should ablate side-chain shortening of the bile-acid precursor. Thus, patients expressing only this variant of ACOX2 (-STL) should present a comparable severity in pathology as patients lacking the enzyme.

D-amino acid oxidase (DAO) is an enzyme that degrades amino acids, and its gene locus was found linked to schizophrenia [34,35,36] along with several links to amyotrophic lateral sclerosis (ALS) [40]. However, it is unclear whether the peroxisomal localization of DAO is linked to these pathologies or whether a cytosolic subfraction of the enzyme performs the relevant activity. This finding renders a clear prediction of the phenotype of patients equipped exclusively with this predominantly cytosolic DAO variant (DAO-S345F) difficult. However, such variant could also serve as an interesting tool to disentangle the relationship between the localization of DAO and its different substrates.

EHHADH functions as enoyl-CoA hydratase/3-hydroxyacyl CoA dehydrogenase. Interestingly, a mutation has been described as being very close to the N-terminus of EHHADH (E3K), which generates a mitochondrial targeting signal in the enzyme [21] and is assumed to interfere with mitochondrial metabolism. Patients with this mutation suffer from renal dysfunction (Inherited Renal Fanconi’s Syndrome) [21,22] which is inherited in an autosomal dominant manner. Thus, the latter model differs from the expected outcome of an SNV affecting the PTS1, because in this case the peroxisomal import is prevented, but no mitochondrial import is expected. In contrast, the phenotype is expected to resemble the mouse model lacking EHHADH, which did not present with any obvious pathology [41], although recently, a defect in the degradation of dicarboxylic fatty acids has been reported [42].

Besides the above peroxisomal proteins that are postulated to lose their PTS1 through the effects of the studied SNVs, we also studied normally cytosolic proteins that have gained a PTS1 motif through SNVs. Four such proteins have been discussed below as they were found to have gene variants potentially generating a PTS1 motif or have increased affinity to the PTS1 receptor after point mutation.

A part of the regulatory subunit of protein phosphatase 4, PPP4R4 is a binding partner of protein phosphatase 4 (PP4) with which it forms a phosphatase specific complex within the cytosol [43] and shares structural similarity with other phosphatase binding proteins such as PR65/A, including several HEAT-motifs [43]. Thus, PPP4R4 might be involved in the regulation of phosphatase activity or target protein selection. PP4 has been linked to diverse physiological functions such as DNA-repair [44], and mice lacking PP4 are embryonically lethal. However, complex formation of PPP4R4 and the core-enzyme (PP4) might even induce a partial peroxisomal co-import of PP4 with PPP4R4 variants harboring a PTS1, which should have even more drastic effects.

Refilin A (RFLNA) is a member of the refilin family (RFLNA and RFLNB), which are filamin-binding short-lived actin regulators coordinating nuclear movements in complex processes such as cell migration and differentiation [45], but also the dynamics of lamellipodium protrusions [46]. Since both refilin proteins are unstable proteins, the consequence of a cytosolic reduction of RFLNA by its mistargeting to peroxisomes is hard to predict. Recently, a homozygous mutation in RFLNA was found in a patient suffering from spondylocarpotarsal synostosis syndrome, a rare syndromic skeletal disorder characterized by disrupted vertebral segmentation with vertebral fusion, carpal and tarsal synostosis, but also with scoliosis and short stature [37].

Rho GTPase activating protein 1 (ARHGAP1) activates the GTPase activity of the Ras-homologue Rho and can be found in the literature under different names (RhoGAP, RhoGAP1, CDC42GAP, and p50rhoGAP). Interestingly, ARHGAP1 has been found as a target of the micro-RNA hs-miR-940, which is spread via exosomes from diverse cancer cells and promotes osteogenic differentiation of human mesenchymal stem cells [47]. Similarly, miR-130b downregulates ARHGAP1 to drive the development of Ewing sarcoma [48], whereas overexpression of ARHGAP1 reduced the proliferation and migration of human cervical carcinoma cells [49]. The physiological relevance of the modulation of ARHGAP1-levels by micro-RNAs suggests that the level of this protein is rather critical, and experimental downregulation of ARHGAP1 causes an increase in tumor marker alkaline phosphatase activity, whereas overexpression of ARHGAP1 caused its reduction [47]. Thus, the level of ARHGAP1 appears critical for the aggressiveness of cancer cells, rendering a dysregulation of its cytosolic level by mislocalization to peroxisomes a prime candidate for further studies.

The hematopoietic prostaglandin D synthase (HPGDS) converts the prostaglandin PGH2 into PGD2 in a glutathione dependent manner [50], but this reaction in vertebrates is performed either by HPGDS (hematopoietic) or PTGDS (lipocalin type). Importantly, the enzyme serves the production of prostanoids in the immune system and mast cells, and its product, PGD2, mediates allergic asthma [51,52,53]. Moreover, the protein is also protective against cerebral ischemia of mice [54] and was specifically found in microglia of mice brains [55]. Finally, an SNV in HPGDS has been linked to the increased probability of developing testicular germ cell tumor [56]. Interestingly, HPGDS forms homodimers [57], and thus, the arbitrary generation of a PTS1 may not only cause mislocalization of this variant, but it might even co-import PTS-less HPGDS engaged in a dimer into peroxisomes. The effect would even further downregulate the cytosolic level of the enzyme, although the specific consequences of downregulation of HPGDS levels in human cells are difficult to predict based on the variety of different functions it performs.

## 3. Discussion

In this study, we examined the possibility of naturally occurring SNVs creating or abolishing a PTS1 through a non-synonymous change in the primary sequence of a protein. Each person’s genome is estimated to contain around 6000 to 10,000 missense variants [58]. While the majority might be inconsequential to health, non-synonymous SNVs resulting in missense mutations can be deleterious to protein function and manifest as severe diseases at the phenotypic level. They have been associated with over 1000 diseases, and the Human Gene Mutation Database (HGMD) currently lists over 100,000 disease-associated missense variants [4]. In this study, in the context of protein localization in the peroxisome, we asked if missense variants found in a sequence variation database could result in the gain or loss of PTS1-mediated peroxisome targeting especially in disease-relevant genes. To this end, we mined gnomAD for SNVs occurring at the PTS1 motif “hotspot” focusing on variants located in the codons encoding the C-terminal tripeptide (Figure 1) of over 30,000 proteins and their isoforms. Over 35,000 unique missense variants were found to cause amino acid substitutions at the tripeptide end for more than 15,000 genes (Figure 2), and we searched for SNVs that possibly led to both gain and loss of function in terms of PTS1-mediated peroxisome localization.

For loss-of-function mutants, the functional consequences of mislocalization of peroxisomal enzymes should be the ablation of enzyme function because essential peroxisomal activity cannot be performed. The now cytosolic enzyme can no longer process educts within the peroxisome, and even if the educts could leave the peroxisomes, the enzyme occurs at a lower concentration in the cytosol and might even require cooperation with one of the peroxisomal enzymes not available in the cytosol to perform its function. In our study, we identified five PTS1-inactivating SNVs within four proteins (Figure 3 and Figure 5A), which should cause a (nearly) complete loss of peroxisomal activity.

Unlike the LoF variants, the concept of disease-causing SNVs due to *de novo* generated targeting signals is new, and it is most interesting for the evaluation of proteins usually found in the cytosol. These proteins should be sensitive to such effects as soluble peroxisomal matrix proteins are transported from the cytosol into the organelle. Moreover, the position of PTS1 at the extreme C-terminus renders its free accessibility much higher than any internal sequence. Although such novel PTS1 are conceptually gain of function (GoF) mutants as they provide a novel faculty to the protein, genetically they should be loss of function because peroxisomal import should sequester the protein from its place of action, whereas the proteins of the other allele should remain in the cytosol. Thus, SNVs of the type described above should become disease-relevant only when both alleles are affected or when the gene dose is critical. However, in a subset of proteins such mutations might even generate a dominant phenotype because oligomeric proteins have been found to become imported in an oligomeric state [59]. Such mechanism has recently been demonstrated for the import of SOD1 [60] and was suggested as a driving force for the co-import of homo-oligomeric protein complexes in which a subfraction of proteins harbor a PTS1 due to reading through a stop codon [61].

In this study, we investigated the effect of annotated SNVs within the C-terminal tripeptide of five human cytosolic proteins, which were predicted to generate a PTS1 (Table 2). In four of these cases (except GLTP), the C-terminus was able to act as a PTS1 and to interact with the receptor PEX5, which suggested that the encoding proteins were also imported into peroxisomes (unless the C-terminus was embedded within the protein) (Figure 4, Figure 5B). None of these SNVs have been identified in a database compiling SNVs associated with diseases (ClinVar) [62]. The observation that none of these SNVs was found in a homozygous state in a database collecting SNVs of healthy (symptom-free) subjects was limited by the low frequency of these mutations.

We also showed that filtering based on a combination of predictions from computational tools, such as the PTS1 predictor and subcellular localization annotation data (see Method), were of value and could be applied to sieve the vast amount of sequence variation data to identify high-quality candidates for further analysis. Specifically, this proved useful when studying GoF variants, because unlike the LoF variants where we could focus on a smaller number of known PTS1-containing proteins; the entire proteome had to be searched to check for *de novo* generation of a PTS1. Filtering in this manner reduced the pool of candidates from over 30,000 to about 16 variants in 16 genes (Appendix A), which would be a manageable size for testing even if all the variants were considered.

As there was some degree of success with using the PTS1-predictor to identify high-quality candidates in the GoF dataset, we decided to assess this tool and another computational method used previously in another study [26]. We sought to determine if they would be useful for future application to newly discovered SNVs to assess their functional outcome on PTS1-mediated transport. Hence, we evaluated the two prediction methods via comparison with the IFA results and the FRET affinity measurements. For the IFA comparison, the PTS1-predictor made correct predictions for ~60% of the tested SNVs, while FoldX had a hit rate of ~70%. Additionally, comparison with affinity measurements demonstrated that both FoldX-computed median ddG_bind_ (Figure 6A) and the PTS1-predictor generated score difference (Figure 6B) had statistically significant correlations with the experimental logarithmic (scaled) ratio of K_a_^app^. The ratio explains the relative reduction or gain in affinity of the point mutant compared to the native protein (log ratio of >0 means that the MT have a lower affinity while if log ratio is <0, MT has greater affinity than the WT). These prediction values may act as a good surrogate estimate for the experimentally-derived ratio and may be beneficial in further prioritizing hits from categorical prediction outcomes of the PTS1 predictor. In addition, we noted that extreme values from these prediction methods could make a distinction between missense variants that induced a change in subcellular localization, and the variants that did not as shown by the segregation of the GoF (in green) and LoF (in red) mutants in the plots from Figure 6. However, further validation with a larger sample set containing a more diverse representation of tripeptide combinations must be performed to determine the robustness of these prediction methods across different situations.

Overall, we demonstrated that sequence variations occurring naturally in the human population have the potential to cause gain and loss in PTS1-mediated peroxisome targeting. Collaboratively, we showed that bioinformatics approaches could be used to search vast amounts of sequence data, and then filter and isolate an experimentally tractable number of candidates for further validation by bench experiments. As the search was limited to the gnomAD database which consists of germline variants depleted for human samples with severe genetic diseases, it would be interesting to perform a similar analysis in a database with somatic variants, such as the Catalogue of Somatic Mutations in Cancer (COSMIC) or The Cancer Genome Atlas (TCGA). Further work will also be required to gain a deeper mechanistic understanding of mistargeting of these proteins and their implications in disease, if at all, and a similar study could be extended to other targeting signals.

## 4. Materials and Methods

### 4.1. Obtaining CCDS Protein Coding Genomic Coordinates and Sequence Information

FASTA files containing protein sequence, DNA sequence, and protein coding coordinates for every CCDS transcript were downloaded from ftp://ftp.ncbi.nlm.nih.gov/pub/CCDS/archive/Hs105/. The relevant data were then extracted and consolidated from three files (“CCDS_protein.current.faa, “CCDS_nucleotide.current.fna” and “CCDS_protein.current.faa”) using a Python (v2.7.13) script. All transcripts with the CCDS status of “Withdrawn” were excluded from further analysis. For each unique protein coding transcript, its corresponding chromosome number and genomic coordinates for the last three amino acid coding codons preceding the stop codon at the C-terminus were obtained to create a query file for subsequent subsetting of gnomAD data.

### 4.2. Parsing gnomAD vcf Files to Obtain Relevant SNV Information

gnomAD v.2.1.1 variant call files (.vcf) for both the Exome and exome-calling intervals of the Genome call sets were downloaded from https://gnomad.broadinstitute.org/downloads along with their corresponding tabix-indexed files (.tbi). The query file from Section 4.1 was used to subset each compressed vcf file separately using tabix v1.7-2 on the Windows Subsystem for Linux (WSL) to obtain all variants occurring at the positions indicated in the query file for each chromosome. A Python script was then created to consolidate, clean, and filter the vcf outputs from tabix. All other variants besides missense variants were excluded from further analysis. For all variants, the Variant Effect Predictor (VEP) annotation tool [63] (v96) was used to acquire variant-specific information such as allele frequencies, ClinVar annotations, and in silico pathogenicity predictions from tools such as PolyPhen2 [64] and SIFT [65].

### 4.3. Obtaining Subcellular Localization Annotations from UniProt

Each unique CCDS transcript from Section 4.1 was mapped to its corresponding UniProt ID, where available, using the UniProt mapping tool available at https://www.uniprot.org/uploadlists/. The mapped UniProt IDs were then used to query the UniProt database and data from the “Subcellular_location_CC” results column was extracted into a tab-delimited file (.tsv) for all queried UniProt IDs. These annotations were then merged with information from Section 4.1 and Section 4.2 with the help of a script utilizing the Pandas library (v0.23.4) in Python. Where available, isoform-specific annotations were retained if a gene had more than one isoform; otherwise the annotation of the canonical transcript of a gene will be extended to all its other isoforms.

### 4.4. Obtaining PTS1 Predictor Results for Native Transcripts and their Corresponding Point Mutants

The PTS1 prediction algorithm developed by Neuberger et al. [25] was used to predict peroxisome localization of native proteins and their variants after SNV induced point mutation. The primary sequence of the protein for each CCDS transcript was consolidated, and the last 12 amino-acid sequences at the C-termini of the transcripts were used as inputs for the prediction algorithm (Metazoa option). The output from the algorithm was then combined with data from Section 4.1, Section 4.2 and Section 4.3 into a single file (Appendix A), and the file was used to search for SNVs relevant to this study.

### 4.5. Prioritization of SNV before Final Selection for Experimental Testing

For the LoF variants, a curated list of 35 gene encoding proteins harboring a PTS1 and containing gnomAD-associated missense variants capable of mutating the PTS1 C-terminal tripeptide region was created (Appendix A). Where available, each gene was annotated with disease annotations from OMIM and genes without these annotations were excluded from further analysis. From the narrowed gene list, further genes were excluded if their products were dually or partly localized in other cell compartments besides the peroxisome based on the UniProt subcellular localization data (see Section 4.3). Specifically, the gene was included in our analysis only if its native product contained the keyword “Peroxisome” but did not have the keywords “Secreted,” “Nucleus,” “Mitochondrion,” “Endosome,” and “Cytoplasm” in its subcellular localization annotation data. Note that at this stage, AGXT (see Appendix A) was included although it had the “Mitochondrion” keyword as the mutant form was annotated to be in the mitochondrion but the native protein is predominantly peroxisomal (Appendix A). All SNVs found in these genes also must pass gnomAD variant quality filters (defined as a “PASS” in the “FILTER” column in either the gnomAD Genome or Exome .vcf call sets). The gnomAD associated quality filters are a collection of hard filters and a random forest classifier model that is applied on each variant to filter out low-quality variants. Some of these filters are based on defined threshold values for parameters, such as the inbreeding coefficient and the genotype quality [28] (further details at https://macarthurlab.org/2018/10/17/gnomad-v2-1/). The variant would be annotated “PASS” if it passes all filters otherwise it would be labeled with the filter that it has failed. To ensure that only high-quality variants likely to be real and not artefactually produced by sequencing errors are studied, all failed variants were not considered for further analysis in this study. After filtering programmatically as described above, we picked only SNVs resulting in mutant tripeptides deviating from the traditional PTS1 motifs [18] in the WT tripeptide motifs for testing of PTS1 ablation.

For the GoF variants, three filters were programmatically applied on the consolidated data from Section 4.3: (i) SNV must pass gnomAD associated quality filters similar to the LoF SNVs, (ii) UniProt subcellular localization annotation of the protein contains the keyword “Cytoplasm” but excludes the keywords “Secreted”, “Nucleus”, “Mitochondrion”, “Endosome”, and “Peroxisome”, and (iii) upon point mutation, PTS1 predictor results must change from “Twilight” or “Not targeted” to “Targeted”. If the SNV fell on a gene with more than one CCDS transcript, the longest transcript was chosen for analysis. After filtering programmatically, we picked 5 SNVs generating a PTS1-like WT tripeptide from non-PTS1 WT tripeptide motif in reasonably long proteins (<1000 amino acids) without transmembrane domains for testing.

### 4.6. FoldX Prediction of Change in Free Binding Energy due to Point Mutations made at PTS1 Ligand-Receptor Interface

A similar procedure as in Reference [26] was used. Using a previously crystallized structure of a PTS1 ligand–PEX5 interaction (pdb ID: 2c0l) [29] as a model, we first truncated the protein-ligand to 12 amino acids in length from the C-terminus. The complex was then energy minimized using the energy minimization function in Yasara Structure (v18.2.7) [66] with the AMBER15FB force field at default settings. The process included minimization of the backbone with short simulated annealing molecular dynamics simulations. Next, the RepairPDB function of the FoldX (v4.0) [33] plugin in Yasara was applied to prepare the complex for FoldX calculation. With the reference ligand being the wild-type protein sequence of each gene, we used the “Mutate residue” option in the FoldX plugin to introduce the point mutant at tripeptide (default settings, 5 runs). We calculated and analyzed the median change in free binding energy (ddG_bind_, kcal/mol) of the complex upon mutation of the ligand at the PTS1 ligand-receptor interface. The illustration of the complex was created using Yasara.

### 4.7. Statistical Analyses

From the affinity measurements of PTS1 quality using FRET, we calculated the log_10_ ratio of relative affinity of the wild-type PTS1 (mean K_app_ WT) over its mutated variant (mean K_app_ MT), given by the formula:(1)$$\log K_{a}^{\;\mathrm{app}}\;\mathrm{ratio}=\log_{10}\;(\frac{\begin{array}{c} K_{a}^{\;\mathrm{app}} \end{array}\;\mathrm{WT}}{\begin{array}{c} K_{a}^{\;\mathrm{app}} \end{array}\;\mathrm{MT}})$$

The log-ratio above was then separately correlated with predicted values from the outputs of the (i) FoldX-computed median ddG_bind_ (kcal/mol) values and (ii) PTS1 predictor (see Section 4.4) score difference (MT Score–WT Score). Correlation analysis (Pearson’s) and hypothesis testing were performed using the base statistical package in R (v3.5.1) and similarly, all graphs were plotted using R.

### 4.8. Cloning

EGFP-fusion proteins: plasmid EGFP-C3 (Clontech) was digested with restriction enzymes BglII and HindIII and then ligated with two oligos of interest (Appendix A) annealed previously. Plasmids were sequenced, and large amounts of DNA were prepared via midi-preps.

### 4.9. Cell Culture

Cells were cultivated in DMEM supplemented with 10% fetal calf serum (FCS) (Sigma-Aldrich, St. Louis, MO, USA), 2 mM L-glutamine, 50 U/mL penicillin, and 100 µg/mL streptomycin ((all Lonza, Basel, Switzerland). The cells were transfected using Turbofect (Thermo-Scientific, Waltham, MA, USA) according to the manufacturer’s instructions using 1 µg DNA and 1.25 µg Turbofect. Then, 40 h after transfection, the cells were fixed for 15 min with 4% paraformaldehyde in PBS (phosphate buffered saline). Cells were washed, permeabilized (5 min with 0.1% Triton X-100 in PBS) and blocked in blocking solution (PBS with 5% FCS and 0.5% bovine serum albumin (BSA, Roche, Basel, Switzerland)). After incubation with primary antibodies (rabbit: α-PMP70; 1:2000, ABR, Golden CO, USA), the slides were washed with PBS several times and exposed to compatible secondary antibodies (Cy3-labelled donkey-α-rabbit IgG, 1:400, Jackson Immuno Research Laboratories, West Grove, PA, USA). Finally, the cells were mounted in PBS/glycerol (1:9) with 3% DABCO (Sigma, St. Louis, MO, USA). For microscopic analysis, an inverted microscope IX71 (Olympus, Wien, Austria) equipped with a CCD camera (CAM-XM10) and appropriate filter sets were used together with C-M-cell software (Olympus, Wien, Austria).

### 4.10. FRET Affinity Measurements

In the approach, we determined the apparent interaction strength between two fluorescent proteins by measuring the energy transfer efficiency between an excited donor molecule (donor) and a proximally located acceptor molecule (acceptor). Thus, we expressed EGFP variants terminating in the different C-terminal PTS1 peptides as donors together with a fusion protein consisting of the PTS1-binding domain of PEX5 (PEX5^TPR^) fused to the C-terminus of mCherry (mCherry-PEX5^TPR^). Utilizing a cell line lacking peroxisomal matrix protein import due to a lack of endogenous full-length PEX5, all the donor proteins remained in the cytosol and their interaction strength with PEX5 could be determined. PEX5^-/-^ cells were prepared at ~70% confluency in 24-well cell culture dishes. The cells were transfected with plasmids bearing the GFP-tagged PTS1 variants and mCherry-tagged PEX5^TPR^ in 2 different ratios (0.4 + 0.6 µg and 0.6 + 0.4 µg), using TurboFect transfection Reagent (Thermo Scientific, Waltham, MA, USA) according to protocol. After incubation for 24 h, the cells were washed in PBS and detached with Trypsin (100 µL per well). Detached cells were transferred to a 96 well v-bottom plate (combining the two different ratios) and then spun for 2 min at 300 g. Supernatant was removed, and the cells were resuspended in 200 µL PBS. The cells were then immediately measured using FlowFRET. FlowFRET was performed on a Cytoflex S flow cytometer (Beckman Coulter, Brea, CA, USA). The donor channel was measured at a 488 nm excitation and 525/40 emission. FRET was measured at a 488 nm excitation and 610/20 emission. The acceptor channel was measured at a 561 nm excitation and 610/20 emission. Data were treated as described in our previous work [32].

## Figures and Tables

**Figure 1 ijms-20-04609-f001:**
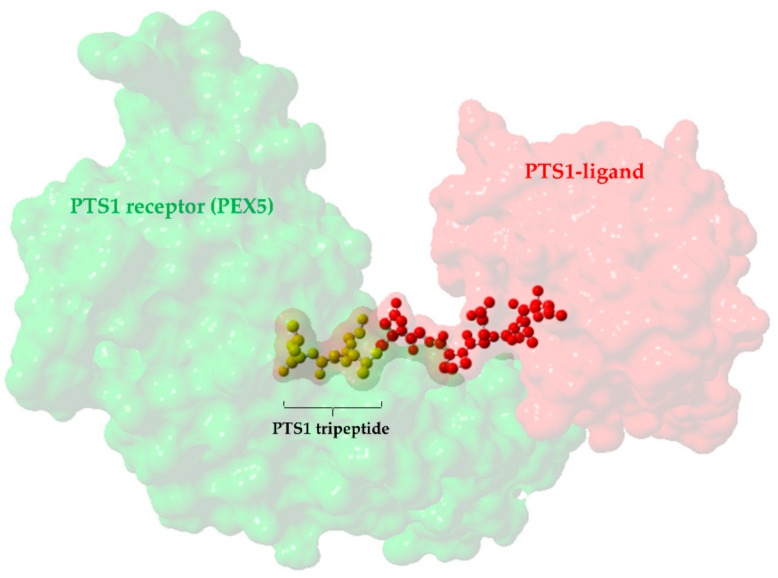
Three-dimensional crystal structure of a PTS1 receptor-ligand complex from Reference [29] (pdb ID = 2c0l), illustrated using Yasara. The PTS1 tripeptide motif (yellow) is located at the extreme C-terminus of the protein (pale red) and is extended and bound to the tetratricopeptide repeat (TPR) region of the receptor (pale green) (C-terminus extension from ligand is highlighted in opaque colors of red and yellow).

**Figure 2 ijms-20-04609-f002:**
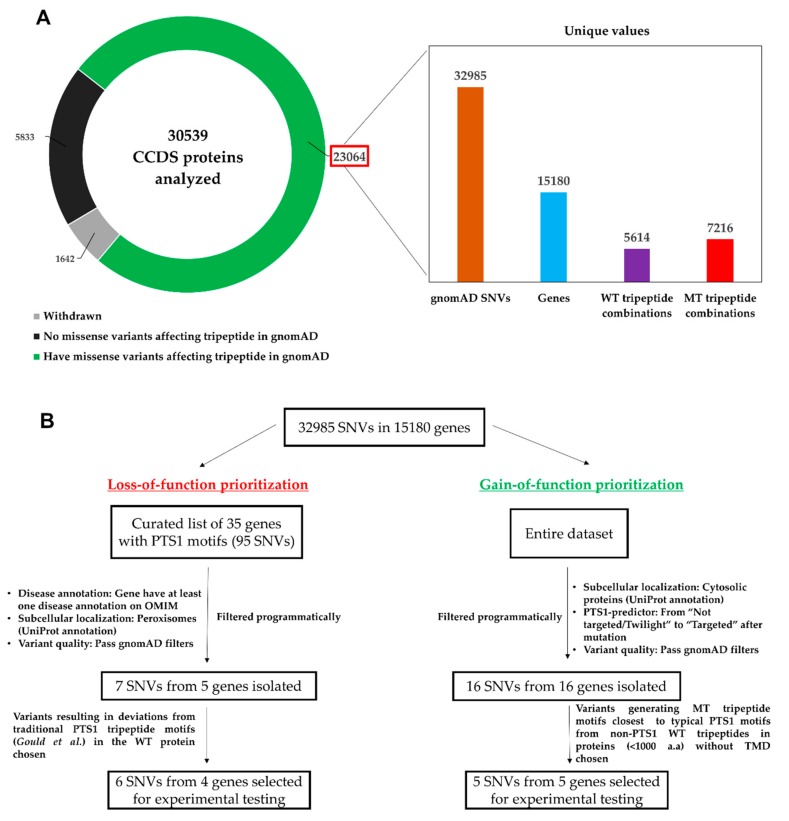
Overall bioinformatics workflow for the mining of sequence variation data from gnomAD and the selection of SNVs for experimental testing. (**A**) 32,985 bi- or multi-allelic gnomAD SNVs capable of altering at least one amino acid in the tripeptide motifs of 23,064 proteins and generating over 7000 unique mutant (MT) tripeptide combinations from 5614 unique wild-type (WT) tripeptides attained (**B**) Prioritization of variants for testing from the SNVs retrieved from gnomAD. TMD, Transmembrane domain; a.a, amino acid.

**Figure 3 ijms-20-04609-f003:**
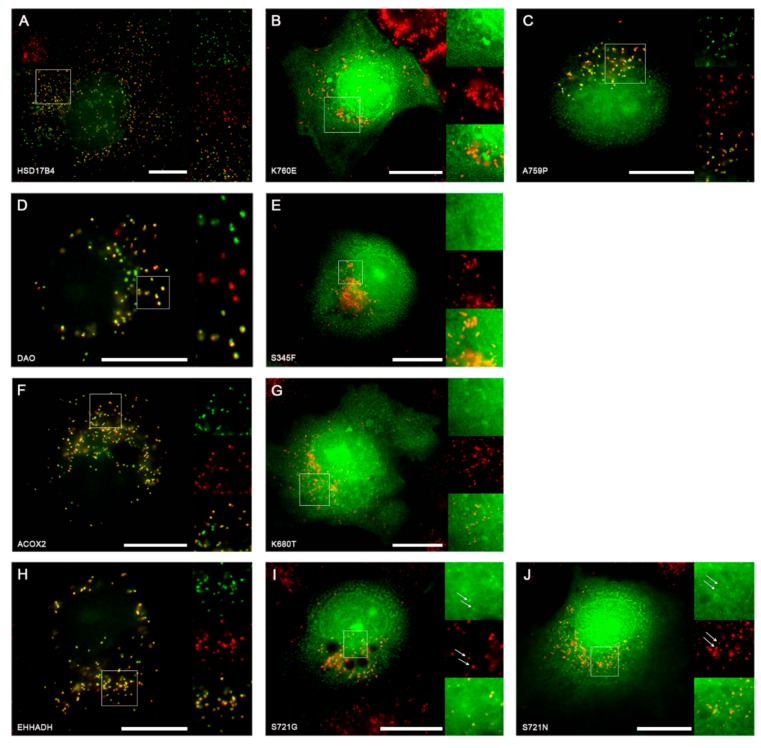
Study of PTS1 loss of function mutations by SNV: (**A**–**J**) COS7 cells were transfected with expression plasmids for different EGFP-PTS1 variants, encoding the C-terminus of multifunctional protein (HSD17B4) (**A**) or the variants K760E (**B**) and A759P (**C**), of d-amino acid oxidase (DAO) (**D**), or the variant S345F (**E**), of acyl-CoA oxidase 2 (ACOX2) (**F**), or the variant K680T (**G**), and of EHHADH (**H**) or the variants S721G (**I**) or S721N (**J**). The subcellular localization was determined by fluorescence microscopy (EGFP, green) in combination with immunofluorescence microscopy of the peroxisomal marker PMP70 (red). **I**,**J**: white arrows indicate co-localization between PMP70 and a small fraction of EGFP. Scale bars indicate 20 µm and white squares define enlarged areas.

**Figure 4 ijms-20-04609-f004:**
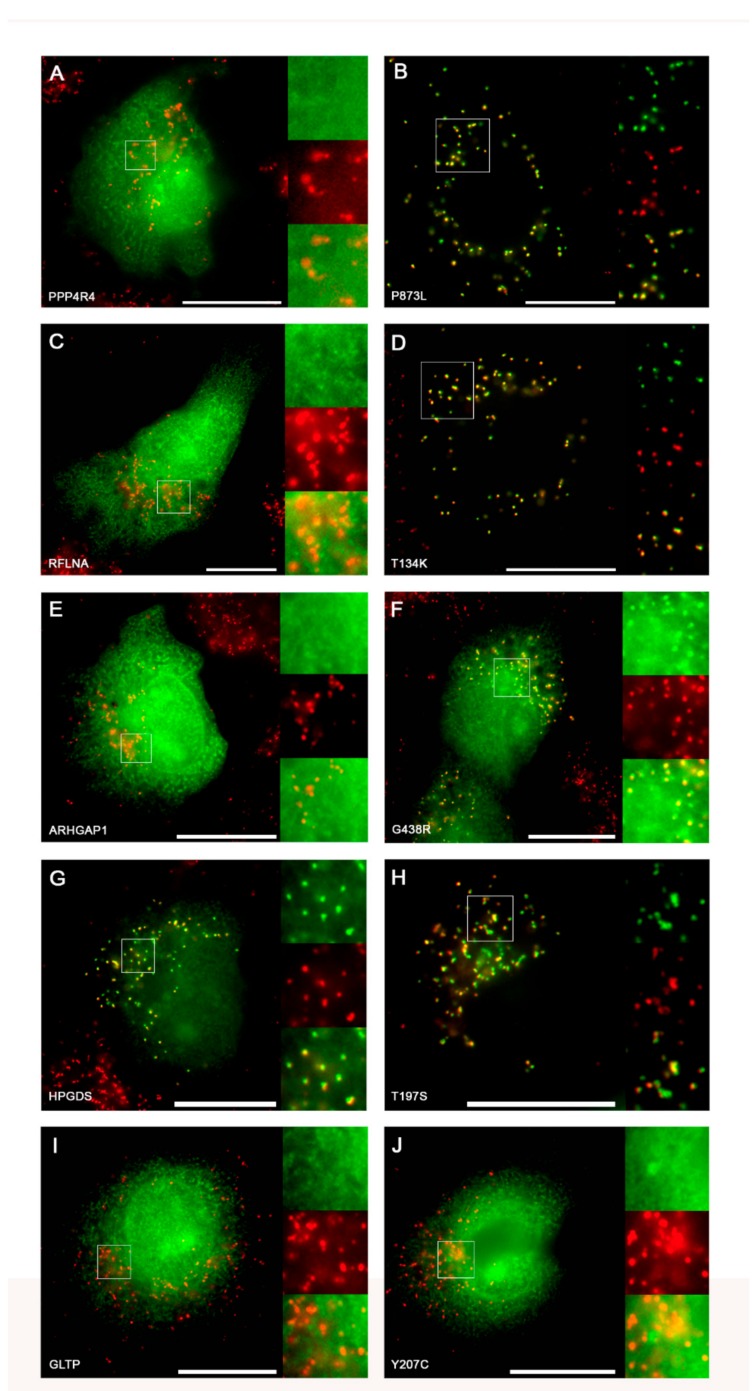
Study of PTS1 gain of function mutations by SNV: (**A**–**J**) COS7 cells were transfected with expression plasmids for different EGFP-PTS1 variants, encoding the C-terminus of PPP4R4 (**A**) or the variant P873L (**B**), of RFLNA (**C**) or the variant T134K (**D**), ARHGAP1 (**E**) or the variant G438R (**F**), of HPGDS (**G**) or the variant T197S (**H**), and of GLTP (**I**) or the variant Y207C (**J**). The subcellular localization was determined by fluorescence microscopy (EGFP, green) in combination with immunofluorescence microscopy of the peroxisomal marker PMP70 (red). Scale bars indicate 20 µm and white squares define enlarged areas.

**Figure 5 ijms-20-04609-f005:**
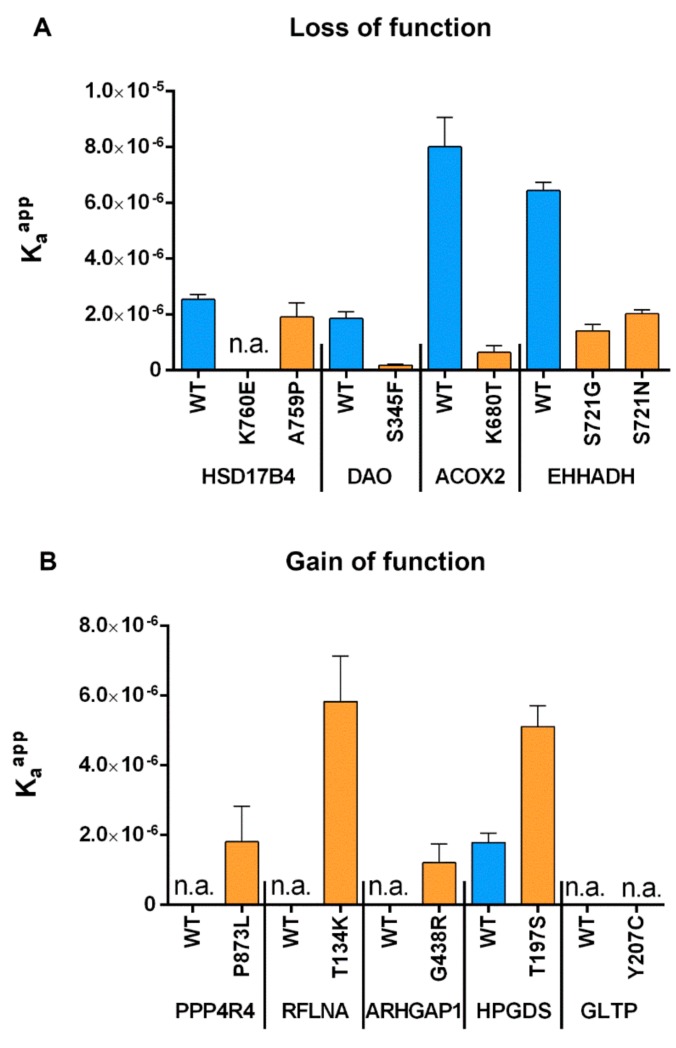
FlowFRET analysis based measurement of affinity between the PTS1 receptor (PEX5) and diverse EGFP-PTS1 variants harboring either the native C-termini of peroxisomal proteins (blue) or LoF mutants thereof (orange) (**A**), or the C-termini of cytosolic proteins (blue) and variants (orange) thereof harboring SNVs (**B**); K_a_^app^: apparent interaction strength as a correlative measure of affinity obtained by fitting; blue: native C-terminus; orange: C-terminus harboring the SNV; n.a: not analyzed (the low affinity of the interaction partners does not allow fitting).

**Figure 6 ijms-20-04609-f006:**
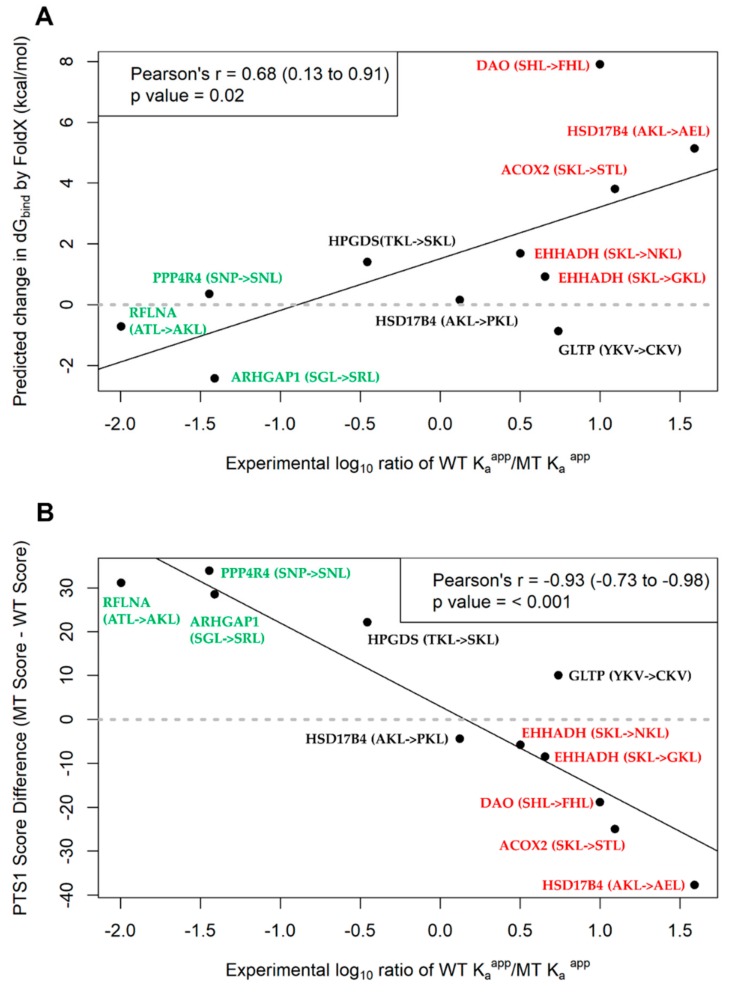
Correlation analysis of the experimental measure of affinity (log_10_ K_a_^app^ ratio) with the prediction values: (**A**) FoldX-computed median ddG_bind_, and (**B**) PTS1 Score difference (MT-WT) upon introducing a single point mutation into the native protein’s tripeptide motif. Proteins and their mutants in green are associated with a GoF change from cytosolic to peroxisomal localization, while mutants in red are associated with a LoF in peroxisome localization. The rest had no change in subcellular localization compartments based on the IFA results. For each plot, we report the Pearson’s *r* values and its associated *p*-value and 95% confidence interval (in brackets).

**Table 1 ijms-20-04609-t001:** SNVs selected for experimental testing for the ablation of PTS1 motif.

rsID. (Nucleotide Change)	Gene	CCDS ID	PTS1 Motif		UniProt Subcellular Localization *
			WT	MT	
rs1049954328 (A>G)	*HSD17B4 ***	CCDS56379.1	AKL	AEL	Peroxisome
rs751064948 (G>C)	*HSD17B4 ***	CCDS56379.1	AKL	PKL	Peroxisome
rs143732132 (C>T)	*DAO*	CCDS9122.1	SHL	FHL	Peroxisome
rs1182979757 (T>G)	*ACOX2*	CCDS33775.1	SKL	STL	Peroxisome
rs138945273 (T>C)	*EHHADH*	CCDS33901.1	SKL	GKL	Peroxisome
rs968361154 (C>T)	*EHHADH*	CCDS33901.1	SKL	NKL	Peroxisome

* Native protein, ** Longest CCDS transcript chosen for analysis. rsID, Reference single nucleotide polymorphism cluster identification; CCDS ID, Consensus Coding Sequence Project accession identification; WT, Wild-type tripeptide; MT, Mutated tripeptide.

**Table 2 ijms-20-04609-t002:** SNVs selected for experimental testing of *de novo* generation of PTS1 motif.

rsID (Nucleotide Change)	Gene	CCDS ID	Motif		PTS1 Prediction (WT)	PTS1 Prediction (MT)	UniProt Subcellular Localization *
			WT	MT			
rs576288488 (C>T)	*PPP4R4*	CCDS9921.1	SNP	SNL	Not targeted	Targeted	Cytoplasm
rs760206157 (C>A)	*RFLNA*	CCDS9258.1	ATL	AKL	Not targeted	Targeted	Cytoplasm, cytoskeleton
rs747965330 (C>T)	*ARHGAP1*	CCDS7922.1	SGL	SRL	Not targeted	Targeted	Cytoplasm
rs765481546 (G>C)	*HPGDS*	CCDS3640.1	TKL	SKL	Not targeted	Targeted	Cytoplasm
rs773190605 (T>C)	*GLTP*	CCDS9136.1	YKV	CKV	Twilight zone	Targeted	Cytoplasm

* native protein. rsID, Reference single nucleotide polymorphism cluster identification; CCDS ID, Consensus Coding Sequence Project accession identification; WT, Wild-type tripeptide; MT, Mutated tripeptide.

**Table 3 ijms-20-04609-t003:** Experimental measurements of apparent affinity strength (K_a_^app^) of the WT and MT PTS1 motif along with predicted values from FoldX (ddG_bind_, kcal/mol) and the PTS1 predictor (score difference, MT-WT) upon motif change.

Gene	Motif		WT K_a_^app^ (10^−6^)	MT K_a_^app^ (10^−6^)	log K_a_^app^ Ratio	FoldX ddG_bind_ (kcal/mol)	PTS1 Predictor Score Difference (MT-WT)
	WT	MT					
*HSD17B4*	AKL	AEL	2.544	0.065	1.592	5.138	−37.8
*HSD17B4*	AKL	PKL	2.544	1.915	0.123	0.154	−4.4
*DAO*	SHL	FHL	1.859	0.186	1.000	7.910	−18.9
*ACOX2*	SKL	STL	8.015	0.646	1.094	3.800	−25.0
*EHHADH*	SKL	GKL	6.444	1.424	0.655	0.920	−8.5
*EHHADH*	SKL	NKL	6.444	2.027	0.502	1.687	−5.8
*PPP4R4*	SNP	SNL	0.065	1.808	−1.444	0.359	33.9
*RFLNA*	ATL	AKL	0.059	5.826	−1.995	−0.723	31.1
*ARHGAP1*	SGL	SRL	0.047	1.209	−1.410	−2.432	28.5
*HPGDS*	TKL	SKL	1.792	5.110	−0.455	1.397	22.2
*GLTP*	YKV	CKV	0.108	0.020	0.738	−0.877	10.1

WT, Wild-type tripeptide; MT, Mutated tripeptide.

**Table 4 ijms-20-04609-t004:** Summary of gene and variant-specific information from gnomAD, OMIM, and ClinVar.

Gene	Associated Disease (OMIM)	rsID	ClinVar	gnomAD Allele Frequency	PolyPhen-2	SIFT
*HSD17B4*	DBP deficiency (type I-III), Perrault Syndrome 1	rs1049954328	NA	Singleton	Possibly damaging	Deleterious
		rs751064948	NA	Singleton	Probably damaging	Deleterious
*DAO*	Schizophrenia *	rs143732132	NA	0.018%, No homozygotes	Possibly damaging	Deleterious
*ACOX2*	Bile acid synthesis defect, congenital	rs1182979757	NA	Singleton	Possibly damaging	Deleterious
*EHHADH*	Fanconi renotubular syndrome 3	rs138945273	Likely benign	0.028%, No homozygotes	Benign	Deleterious
		rs968361154	NA	Singleton	Benign	Tolerated
*PPP4R4*	None	rs576288488	NA	0.009%, No homozygotes	Benign	Deleterious **
*RFLNA*	None	rs760206157	NA	0.003%, No homozygotes	Benign	Tolerated
*ARHGAP1*	None	rs747965330	NA	<0.001%, No homozygotes	Probably damaging	Deleterious **
*HPGDS*	None	rs765481546	NA	Singleton	Possibly damaging	Deleterious
*GLTP*	None	rs773190605	NA	Singleton	Possibly damaging	Deleterious

* Association awaiting confirmation on OMIM, ** Low confidence prediction. rsID, Reference single nucleotide polymorphism cluster identification; gnomAD, Genome Aggregation Database.

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
