# Peer review of "Rare Human Missense Variants can affect the Function of Disease-Relevant Proteins by Loss and Gain of Peroxisomal Targeting Motifs"

_ijms, 2019, doi:10.3390/ijms20184609_

Round 1
Reviewer 1 Report
Chong and colleagues filtered the Genome Aggregation Database for entries that result in the loss or gain of a peroxisomal targeting sequence on the encoded protein. After identifying 6 and 5 candidate sequences for aforementioned cases, respectively, they investigated the effect of the variation by 1) localisation studies of each eGFP-PTS1 fusion protein in COS7 cells, and 2) by a FRET based approach, determining the relative binding affinity of the eGFP-PTS1 protein to Pex5. Finally, they use the results of the two approaches to evaluate the performance of the PTS prediction programs PTS1 predictor and FoldX, which had hit rates on the 11 tested sequences of 60% and 70%, respectively.
The manuscript is well written and mostly easy to follow. The localisation studies and binding affinity measurement using FRET match in 7 of the 11 cases investigated. However, it is difficult to understand how a specific binding affinity or peroxisomal localisation of PTS1-EGFP is related to the potential of the variation to cause a disease. Therefore, the reader is left wondering, how to assess the importance of the presented data. Analysis of pathogenic mutations shows that the genetic background of a mutation strongly influences its phenotypic severity. In contrast, some of the investigated variations might have such a strong impact on peroxisomal or cellular function that fertilised eggs would not survive to delivery. Predicting disease from isolated genetic variations thus seems far-fetched. A clear statement in the introduction as to how to assess the importance of the presented data might help understand the purpose of the analysis. It would be beneficial to add some statistics on the ~33000 identified missense variants. Are there any variants in this pool that actually are associated with a disease? How is the distribution of variations over all three residues? Are there specific variations that occur more often? Also, the contribution of sequencing errors to the entries in the gnomAD should be discussed. Besides the questions mentioned above, the authors should consider the following remarks:
Major points:
The procedure to select the 5 GoF and 6 LoF variants is incomprehensible. Figure 2 indicates that prioritization was done based on expert opinion, disease, subcellular localisation, and PTS1 predictor results. The materials and methods section does not explain in detail how the 32985 targets were narrowed down to 5-6 targets. Figure 2 needs to be extended by the steps taken to get from 32985 targets to 5-6 targets. More controls should be shown in the FRET analysis in figure 5. The Kaapp values are very small. A positive control of a EGFP-PTS1-Pex5-mCherry fusion protein as well as a negative control of individually expressed EGFP and mCherry should be shown. The correlation analysis between experimental data and prediction values is obsolete if the selection of the target sequences was done based on one of the prediction programs (PTS1 predictor) as stated in Figure 2.Minor points:
- Line 63: the peroxisomal targeting sequence should be introduced as S/A/C - K/R/H - L -COO-
- Line 116: should read "These gene products must..."
Author Response
Reviewer 1
Comment 1:
However, it is difficult to understand how a specific binding affinity or peroxisomal localisation of PTS1-EGFP is related to the potential of the variation to cause a disease. Therefore, the reader is left wondering, how to assess the importance of the presented data. Analysis of pathogenic mutations shows that the genetic background of a mutation strongly influences its phenotypic severity. In contrast, some of the investigated variations might have such a strong impact on peroxisomal or cellular function that fertilised eggs would not survive to delivery. Predicting disease from isolated genetic variations thus seems farfetched.
A clear statement in the introduction as to how to assess the importance of the presented data might help understand the purpose of the analysis.
Response 1:
We understand the reviewer’s concern and we are not suggesting that mutations in this study can solely predict or cause disease which we have clarified in the introduction as suggested and as indicated below.
But rather, our study suggests the potential of inter-individual variation to cause aberrant protein localization and that this an important but often overlooked aspect when assessing the pathogenicity of point mutants.
For causal prediction of disease, as rightly pointed out, this should be one but not the only aspect considered. At the protein level, correctly predicting such changes in peroxisomal localization will allow us to discuss their implications on the protein’s cellular and molecular function and compare them with known
phenotypic presentations if the protein have been associated with a disease (eg: HSD17B4 and ACOX2).
Line 88-91: “Our findings would be important in the context of analyzing the effects of variant-induced
aberrant localization on protein function and reinforces the need to evaluate targeting signal changes when
determining the disease-relevance of a protein mutation amongst other factors.”
Comment 2:
It would be beneficial to add some statistics on the ~33000 identified missense variants. Are there any variants in this pool that actually are associated with a disease?
Response 2:
As requested by the reviewer, while we could not manually curate the entire set of over 30,000 SNVs to identify ones that have led to disease, we nevertheless gathered disease annotations for these variants from ClinVar, where available, using the variant annotation tool (VEP v96). In doing so, we found 29 SNVs
that have at least one “pathogenic” or “likely pathogenic” clinical significance reported in ClinVar. As expected, most of these variants are heterozygotic just like the SNVs investigated by us, except for two of them: rs1800456 C>A which is linked with ornithine aminotransferase deficiency (Doimo M et al. Human
Mutation 2013) and rs139596860 T>A associated with X-linked protoporphyria (Balwani et al. GeneReviews 2013) which are present in homozygous carriers in gnomAD. We have added the disease annotation statistic from ClinVar to the paper (as per below) and have updated the Methods section 4.2.
We thank the reviewer for bringing up this pertinent point.
Lines 339-343: “In the ClinVar database, we found the most frequent variant S721G (in EHHADH) to be classified as “likely benign” while all other SNVs were not listed although if we consider the entire collection of missense variants, 29 of them have been associated with at least one “pathogenic” or “likely pathogenic” clinical significance in this database.”
Comment 3:
How is the distribution of variations over all three residues? Are there specific variations that occur more often?
Response 3
In the case of variation over the last three positions, we found that all three positions are equally likely to have mutations. Specifically, these SNVs result in 16309 mutations at the last amino acid (-1 position), 16453 mutations at the -2 position and 16681 mutations at the -3 position in the proteins studied. As for
specific variations, in our dataset, there were 155 unique pairs of mutations with the most common mutation in the tripeptide being from proline (WT) to leucine (MT) which occurred 1365 times against an average of 319 times per pair. But comparison with Marharyta et al. (Human Mutation 2015) who analyzed the
frequency of mutation combinations in the HumVar database suggests that the P->L is one of the more common mutations and this is unlikely an enrichment specific to this dataset.
Comment 4:
Also, the contribution of sequencing errors to the entries in the gnomAD should be discussed.
Response 4:
With regards to sequencing error, the gnomAD authors aggregated data from over 30 sequencing studies (listed here: https://gnomad.broadinstitute.org/about) after which they harmonized and summarized the variant information collected. Hence, it would be hard to pinpoint a specific error rate as it would be
dependent on the sequencing technology used in each study but if we assume that most of the studies used short read NGS technology, it has been reported to have an error rate of 10-3 (Fox, et al Next Gener Seq Appl 2015). We understand the concern that a variant may be artefactually produced by these
sequencing errors and hence might not be a “real” variant even though it is listed in the database. To allay such concerns, all variants tested in our study have passed the variant quality filters imposed by gnomAD and we have excluded all variants that do not pass them even if they might be promising. Ultimately, this boosts the chances that the variant might be real but is not a 100% guarantee as quality assessment of variants is probabilistic (in gnomAD they used a machine learning model to classify variants) and remains
a challenge. We have added more information regarding the variant QC process from gnomAD in the Methods section 4.5 to provide more details (indicated below) on how we have performed the filter above.
Lines 558-565: “All SNVs found in these genes also must pass gnomAD variant quality filters (defined as a “PASS” in the “FILTER” column in either the gnomAD Genome or Exome .vcf callsets). The gnomAD associated quality filters are a collection of hard filters and a random forest classifier model that is applied
on each variant to filter out low quality variants. Some of these filters are based on defined thresholds values for parameters such as the inbreeding coefficient and the genotype quality (further details at https://macarthurlab.org/2018/10/17/gnomad-v2-1/). The variant would be annotated “PASS” if it passes all
filters otherwise it would be labelled with the filter that it has failed. To ensure only high-quality variants likely to be real and not artefactually produced by sequencing errors are studied, all failed variants were not considered for further analysis in this study.”
Comment 5:
The procedure to select the 5 GoF and 6 LoF variants is incomprehensible. Figure 2 indicates that prioritization was done based on expert opinion, disease, subcellular localisation, and PTS1 predictor results. The materials and methods section does not explain in detail how the 32985 targets were narrowed
down to 5-6 targets. Figure 2 needs to be extended by the steps taken to get from 32985 targets to 5-6 targets.
Response 5:
We acknowledge that the previous figure was unclear and to improve the clarity of the selection procedure, we have replaced Figure 2 with two figures (Figure A1 previously from the Appendix and a new figure) that explicitly details the entire selection process for both the LoF and GoF variants. We have also updated the Methods section 4.5 with additional information and included a new table in the Appendix to list the 35 genes curated for the LoF prioritization (Table A1). For the GoF variants, the rsIDs of the 16 variants from 16 genes is available in Table A2 of the Appendix. And finally, the entire dataset is also available in the Appendix as Table A3.
Comment 6:
More controls should be shown in the FRET analysis in figure 5. The Kaapp values are very small. A positive control of a EGFP-PTS1-Pex5-mCherry fusion protein as well as a negative control of individually expressed EGFP and mCherry should be shown.
Response 6:
We thank the reviewer for his comment and agree that the resulting numbers for Ka app might appear unintuitively low, but they originate from the applied FRET-method, which has some specificities described in our recent paper Hochreiter, et al (Sci.Reports 2019).
Please note that the presented values for apparent interaction strength (Ka
app) are results of a fitting algorithm and are related to the real measure of interaction strength, Ka, by a proportionality factor, which originates from the normalization procedure but cannot be determined directly. However, this factor affects all measurements to the same extent and thus any change in affinity can be delineated from results obtained by measurement and fitting of different PTS1 peptides. Numerically, this proportionality factor is related the
concentration of interacting molecules inside the cell (to calculate Ka) and intensity measures reflecting the amount of these proteins as obtained by the detector. Unfortunately, the reference values suggested cannot be offered by this approach, because for non-interacting proteins FRET-efficiency values (DFRET) do not differ from the background and thus cannot be subjected to fitting, whereas fusion proteins such as the suggested EGFP-PTS1-Pex5-mCherry should display infinite affinity, but its determination just results in
infinitely large confidence intervals (as demonstrated in Hochreiter, et al; Sci.Reports (2019)).
To support this explanation, we added an additional figure for the reviewer below, which shows the DFRET curves for the fusion protein mCherry-EGFP (green), for co-expressed non-interacting proteins mCherry and EGFP (red) and for mCherry-PEX5TPR and EGFP-EGFP (HSD17B4) (blue). In [a] the DFRET-values of all cells are plotted against the acceptor (mCherry) to donor (EGFP) ratio and in [b] the average DFRETvalues of cells sharing a near equimolar ratio are plotted for the fusion protein (green), the separately expressed EGFP and mCherry proteins (red) and the different combinations of mCherry-PEX5TPR and
EGFP-PTS1 variants. However, as these DFRET values do not directly reflect changes in affinity we do not intend to include these data into the paper.
Comment 7:
The correlation analysis between experimental data and prediction values is obsolete if the selection of the target sequences was done based on one of the prediction programs (PTS1 predictor) as stated in Figure 2.
Response 7:
Thank you for pointing this out. While we understand how the reviewer might have come to this conclusion, we would like to highlight that the selection of mutants using the PTS1 predictor was conducted using the crude label outputs from the predictor (“Not targeted”/” Twilight” to “Targeted”) and not by absolute score differences between the WT and MT sequences and even then, this was only applied for the GoF mutants. Numerically, filtering using the labels this way amounted to finding mutations resulting in a score difference of more than 0 where all score differences above zero are considered positive – that is, a change of localization/GoF is predicted to happen. In our analysis in Figure 6B, we ask a different question: beyond using the crude labels which assumes all score differences above 0 as positive, is there meaning in the absolute differences generated and if so, can they then serve as a finer differentiating point between two or more mutations with the same crude label outcomes.
Additional clarification, In addition to the above, all minor changes suggested have also been added to the manuscript and we thank the reviewer the comments on the good writing quality and readability of the paper.
Sincerely,
Cheng-Shoong Chong, Markus Kunze, Sebastian Maurer-Stroh

Reviewer 2 Report
The manuscript ‘Rare human missense variants can affect function of disease relevant proteins by loss and gain of peroxisomal targeting motifs’ by Chong, Kunze, et al. analyses how single nucleotide variants can lead to the gain and loss of PTS1 (peroxisome targeting signal type 1). Altogether this is a novel and nice study that should be published with minor changes.
p2l79 – gnomAD should briefly be introduced to the reader. How complete is it? Where does the data come from? Are there other databases of a similar scope? (Why did the authors chose this one?) p3l96 – “As illustrated in Figure 1 we chose to focus on SNVs … ” – Correspondence with figure 1 should be improved. Inhowfar does Figure 1 illustrate the choice? Why do authors concentrate only on the PTS1 tripeptide? p3l105 Figure 1 – The purpose/function of this figure is not clear to me. The authors should also explain how the figure was drawn. What is shown in red as opposed to pale red? Which PTS1 ligand is shown? Is all of PEX5 drawn in the figure (as indicated in the figure), or only the TPR (as indicated in the legend)? p4109 Figure 2 and Text p4: Figure 2 and supplementary Figure A1 should be merged and the selection criteria should be more clearly indicated. How are “relevant” transcripts defined? What are “gnomAD associated quality filters”? Can the prioritization steps be more clearly defined? p4l121 – Table 1 – Abbreviations (rsID., MT) should be explained. p3l140 Figure 3 and p7l180 Figure 4. Size bars are needed in both figures! Authors should mark the origin and the purpose of the insets! The figures would be more easily readable if some of the labels from the legend could be directly indicated in the figure (gene names, variants). p5l148 – Section 2.3.1 – I suggest to make the selection process of “de novo PTS1” more transparent. Currently the authors refer to Methods 4.5. and “strict filters” – however the process is not entirely transparent. Authors should list all 35 genes in Table A1 and indicate which gene passes which filter to arrive at 16 SNVs and then at 5 SNVs. p6l23 – ‘minor gain of function’. Does the prediction/targeting of the WT affect the prediction/targeting of the mutant? The authors somewhat suggest that an ambivalent WT prediction/targeting can lead to a less confident mutant prediction/targeting! This would not be logical, because one tripeptide does not know the other! The authors should rather discuss the amino acids in the tripeptide (a ‘poor’ V in position -1, and a change from Y to C in position -3). This point is related to the following: p10l220 Table 3, p12 Figure6B, and Text: In the light of what was said before, that authors should more clearly define and discuss why and under which circumstances it could be useful to calculate the predictor score difference. In the light of the previous point, it may be a concept of limited value. I understand that it may be of interest to compare this value to Kaapp and FoldX values, but this should be done more stringently. Inherent to the concept of predictor score difference is the fact that a SNV can only change one position in the PTS, so there may be a general threshold (for both gof and lof). I guess this is more relevant for the gof, because any very good PTS1 can always be rendered non-functional by a SNV, whereas a single nucleotide change could not render any non-PTS1 into a PTS1. p9 Figure 5. I suggest indicating values on the y-axix in as ‘10e-6’ and to subgroup the data that belongs to a given gene on the x-axis (by dotted lines or changes in spacing). p12l273 check spelling “Pearson’s” p16-19 Materials and methods: check use of comma / point in numbers (eg. p18l534: 0.5 instead of 0,5), spacings (eg. 100 µl instead of 100µl, 561 nm instead of 561nm), spelling (p18l539 'inverted microscope' instead of 'invert microscope'). p19l575 change “oligomers” to “oligonucleotides”Altogether I enjoyed reading the ms. very much, and the points listed above are only meant to improve the clarity and readability of the paper.
Author Response
Reviewer 2
Comment 1:
p2l79 – gnomAD should briefly be introduced to the reader. How complete is it? Where does the data come from? Are there other databases of a similar scope? (Why did the authors chose this one?)
Response 1:
Thanks for the suggestion. We agree and have added to the manuscript more details about gnomAD and the reason behind choosing it for our analysis.
p3|103-107: “gnomAD is a sequence variation database containing harmonized variant data from more than 140,000 human samples collected from a broad range of studies (full list available at: https://gnomad.broadinstitute.org/about). We chose this database for our analysis as it is currently, the largest publicly available human sequence variation database and the data is easily accessible”
Comment 2:
p3l96 – “As illustrated in Figure 1 we chose to focus on SNVs … ” – Correspondence with figure 1 should be improved. Inhowfar does Figure 1 illustrate the choice? Why do authors concentrate only on the PTS1 tripeptide? p3l105 Figure 1 – The purpose/function of this figure is not clear to me. The authors should also explain how the figure was drawn. What is shown in red as opposed to pale red? Which PTS1 ligand is shown? Is all of PEX5 drawn in the figure (as indicated in the figure), or only the TPR (as indicated in the legend)?
Response 2:
Thanks for pointing out the discordance between the manuscript text and the legend for Figure 1. To improve clarity, we have reworded the text and the legend to provide more details about Figure 1 including explanations for the color scheme and the method we used to draw the Figure. The intention of Figure 1 is to highlight the location of the PTS1 tripeptide motif in the ligand using a previously crystallized PTS1-receptor (TPR region) and ligand complex as an example. We have also addressed our choice to concentrate on tripeptide motif in PTS1 in the manuscript. The change in the text is indicated as per below:
p3|109-115: “Additionally, we chose to focus on SNVs lying in the last three codons of each transcript as they can potentially generate or abolish a PTS1 mediating the interaction with the PTS1 receptor (PEX5) by mutating the C-terminal tripeptide of the proteins analyzed. The importance of this tripeptide to PTS1-mediated peroxisome targeting has been extensively studied as compared to its upstream sequence and hence, is well suited to predict and proof functional changes in the PTS1. Figure 1 depicts the location of the tripeptide motif in PTS1 in a previously crystallized complex (pdb ID: 2c0l) that consists of the TPR region of PEX5 and its PTS1-containing ligand (human SCP2).”
Comment 3:
p4109 Figure 2 and Text p4: Figure 2 and supplementary Figure A1 should be merged and the selection criteria should be more clearly indicated. How are “relevant” transcripts defined? What are “gnomAD associated quality filters”? Can the prioritization steps be more clearly defined?
Response 3:
As suggested, we have improved the clarity of Figure A1 (now Figure 2A) by redefining “relevant” transcripts as CCDS proteins with missense variants from gnomAD affecting the tripeptide motif. To better define the
prioritization steps for both the LoF and GoF selections, we have replaced the old Figure 2 with a new figure (Figure 2B) which clearly defines the steps taken and the outcome at each step. Additionally, we have also added more information on gnomAD associated quality filters in the Methods section 4 (as indicated in response 4).
Comment 4:
p5l148 – Section 2.3.1 – I suggest to make the selection process of “de novo PTS1” more transparent. Currently the authors refer to Methods 4.5. and “strict filters” – however the process is not entirely transparent. Authors should list all 35 genes in Table A1 and indicate which gene passes which filter to arrive at 16 SNVs and then at 5 SNVs.
Response 4:
As per the Response in comment 3, we have explicitly summarized the selection process in Figure 2B and have also included additional information (as Table A1) regarding the 35 genes analyzed for prioritization of the loss-of-function (LoF) mutants including their subcellular localization and disease annotation data. The Methods section 4.5 has also been updated with additional information regarding the selection process as also indicated below.
p23|544-566 (LoF mutants): “For the LoF variants, a curated list of 35 genes encoding proteins harboring a PTS1 and containing gnomAD-associated missense variants capable of mutating the PTS1 C-terminal tripeptide region was created (Table A1). Where available, each gene was annotated with disease annotations from OMIM and genes without these annotations were excluded from further analysis. From the narrowed gene list, further genes were excluded if their products were dually or partly localized in other cell compartments besides the peroxisome based on UniProt subcellular localization data (see section 4.3). Specifically, the gene was included in our analysis only if its native product contained the keyword “Peroxisome” but do not have the keywords “Secreted”, “Nucleus, “Mitochondrion”, “Endosome” and “Cytoplasm” in its subcellular localization annotation data. Note that at this stage, AGXT (see Table A1) was included although it had the “Mitochondrion” keyword as the mutant form is annotated to be in the mitochondrion, but the native protein is predominantly peroxisomal (Table A1). All SNVs found in these genes also must pass gnomAD variant quality filters (defined as a “PASS” in the “FILTER” column in either the gnomAD Genome or Exome .vcf callsets). The gnomAD associated quality filters are a collection of hard filters and a random forest classifier model that is applied on each variant to filter out low quality variants. Some of these filters are based on defined thresholds values for parameters such as the inbreeding coefficient and the genotype quality (further details at https://macarthurlab.org/2018/10/17/gnomad-v2-1/). The variant would be annotated “PASS” if it passes all filters otherwise it would be labelled with the filter that it has failed. To ensure only high high-quality variants likely to be real and not artefactually produced by sequencing errors, all failed variants were not considered for further analysis in this study. After filtering programmatically as described above, we picked only SNVs resulting in mutant tripeptide deviating from the traditional PTS1 motifs in the WT tripeptide motifs for testing of PTS1 ablation.
p23|567-575 (GoF mutants): “For the GoF variants, three filters were programmatically applied on the consolidated data from section 4.3: i) SNV must pass gnomAD associated quality filters similar to the LoF SNVs, ii) UniProt subcellular localization annotation of the protein contains the keyword “Cytoplasm” but excludes the keywords “Secreted”, “Nucleus, “Mitochondrion”, “Endosome” and “Peroxisome”, iii) Upon point mutation, PTS1 predictor results must change from “Twilight” or “Not targeted” to “Targeted”. If the SNV fell on a gene with more than one CCDS transcripts, the longest transcript was chosen for analysis. After filtering programmatically, we picked 5 SNVs generating a PTS1-like WT tripeptide from non-PTS1 WT tripeptide motif in reasonably long proteins (< 1000 amino acids) without transmembrane domains for testing.”
Comment 5:
p6l23 – ‘minor gain of function’. Does the prediction/targeting of the WT affect the prediction/targeting of the mutant? The authors somewhat suggest that an ambivalent WT prediction/targeting can lead to a less confident mutant prediction/targeting! This would not be logical, because one tripeptide does not know the other! The authors should rather discuss the amino acids in the tripeptide (a ‘poor’ V in position -1, and a change from Y to C in position -3).
Response 5:
We acknowledge that the intention of the sentence is unclear, and the prediction is merely an observation made and was not meant to be used to justify the lack of GoF observed for the GLTP YKV->CKV mutation. We have since rephrased the sentence as indicated below and hope this clarifies our intention. Additionally, we agree with the author’s suggestion and have added the discussion into the manuscript.
p9|217-221: “However, in this case the mutation increased the numerical PTS1-score only slightly, although this was sufficient to cross a threshold and change the qualitative prediction from “Twilight zone” to “Targeted”. The inability of this mutant to act as a PTS1 might be due to the valine being the last residue of the tripeptide as this amino acid has been shown to weaken known PTS1 motifs.”
Comment 6:
This point is related to the following: p10l220 Table 3, p12 Figure6B, and Text: In the light of what was said before, that authors should more clearly define and discuss why and under which circumstances it could be useful to calculate the predictor score difference. In the light of the previous point, it may be a concept of limited value. I understand that it may be of interest to compare this value to Kaapp and FoldX values, but this should be done more stringently. Inherent to the concept of predictor score difference is the fact that a SNV can only change one position in the PTS, so there may be a general threshold (for both gof and lof).
Response 6:
We agree with the reviewer that the rationale for the score difference should be more prominently stated in the manuscript and have added the necessary clarification as per below. The usefulness of calculating predictor score differences arise in a large dataset when there are multiple pairs of WT and MT protein with the same crude label outcomes. As an example, in our dataset there are 163 entries where upon mutation from WT to MT tripeptide the classification changes from “Not targeted” to “Targeted”. Hence, in this instance and other similar examples (both LoF and GoF), we ask if the score difference calculated can serve as a finer differentiation point to further subdivide and prioritize the most relevant mutations for experimental testing. Its use however, needs to be validated and in this study as pointed out by the reviewer, we compared them against the experimental affinity measures and found that they correlated well (Figure 6B) and might serve as a proxy for these measurements. In other words, we ask: rather than just predict the localization change could the tool be useful as a quantitative indicator of affinity changes if we consider the score difference? A caveat however as discussed in the paper is that more samples including greater diversity of tripeptide combinations will be needed for further validation before this can be implemented.
p16|315-317: “Since this method is sequence-based and the calculation is fast also over a larger number of sequences (<1 sec per sequence), it seems to be well suited for high-throughput screening of effects of SNVs on PTS1 motifs especially in large datasets where the score difference could serve as a finer differentiation point for mutations resulting in the same pairs of categorical outcomes (eg “Targeted” -> “Not targeted”)”
p22|487-488: “…these prediction values may act as a good surrogate estimate for the experimentally-derived ratio and may be beneficial in further prioritizing hits from categorical prediction outcomes of the PTS1 predictor”
Comment 7:
I guess this is more relevant for the gof, because any very good PTS1 can always be rendered non-functional by a SNV, whereas a single nucleotide change could not render any non-PTS1 into a PTS1.
Response 7:
We see the reviewer’s point but disagree with the assumption that a SNV could not generate a non-PTS1 into a PTS1, because this appears as a reverse process as from PTS1 to non-PTS1, which is considered possible by the reviewer and is demonstrated by the gain of function mutants in our manuscript. However, we agree that additional restrictions outside of the C-terminal PTS1 such as 3D structure of the protein might put additional limitations to the generation of de novo peroxisomal proteins (more so than for loss of function as pointed out by the reviewer) although the predictor tries to grasp properties such as exposition of the C-terminus and is hence, useful for both cases (LoF and GoF).
Comment 8:
p3l140 Figure 3 and p7l180 Figure 4. Size bars are needed in both figures! Authors should mark the origin and the purpose of the insets! The figures would be more easily readable if some of the labels from the legend could be directly indicated in the figure (gene names, variants).
Response 8:
We are grateful for the reviewer’s comment. We have marked the enlarged areas in figures 3 and 4, scale bars have been introduced and figure legends have been adapted accordingly. In the course of this procedure we found an incorrect enlargement in Fig.4H, which now has been corrected.
Additional clarification
In addition to the above, all other typographical errors, lack of abbreviation explanations and suggestions for improvements to figures pointed out by the reviewer have been corrected or added.
We thank the reviewer as well for the kind words and praise for the novelty of the paper and we are glad that the reviewer enjoyed reading the paper.
Sincerely,
Cheng-Shoong Chong, Markus Kunze, Sebastian Maurer-Stroh
Round 2
Reviewer 1 Report
I am satisfied by the answers of the authors to my comments. The revisions have improved the manuscript and strengthened their conclusions. They also helped me to better understand their findings.